# Beyond Isolated Facts: Synthesizing Narrative and Grounded Supervision for VideoQA

## Abstract

The performance of Video Question Answering (VideoQA) models is fundamentally constrained by the nature of their supervision, which typically consists of isolated, factual question-answer pairs. This "bag-of-facts" approach fails to capture the underlying narrative and causal structure of events, limiting models to a shallow understanding of video content. To move beyond this paradigm, we introduce a framework to synthesize richer supervisory signals. We propose two complementary strategies: Question-Based Paraphrasing (QBP), which synthesizes the diverse inquiries (what, how, why) from a video's existing set of question-answer pairs into a holistic narrative paragraph that reconstructs the video's event structure; and Question-Based Captioning (QBC), which generates fine-grained visual rationales, grounding the answer to each question in specific, relevant evidence. Leveraging powerful generative models, we use this synthetic data to train VideoQA models under a unified next-token prediction objective. Extensive experiments on STAR and NExT-QA validate our approach, demonstrating significant accuracy gains and establishing new state-of-the-art results, such as improving a 3B model to 72.5% on STAR (+4.9%) and a 7B model to 80.8% on NExT-QA. Beyond accuracy, our analysis reveals that both QBP and QBC substantially enhance cross-dataset generalization, with QBP additionally accelerating model convergence by over 2.5x. These results demonstrate that shifting data synthesis from isolated facts to narrative coherence and grounded rationales yields a more accurate, efficient, and generalizable training paradigm.

## 1 Introduction

Video Question Answering (VideoQA) (Patel et al., 2021; Zhong et al., 2022) is a pivotal multimodal task that requires models to reason over complex visual and textual inputs to answer natural language questions about videos. It has broad applications, including video retrieval and surveillance (Sreenu & Durai, 2019), as well as assistive technologies and interactive AI systems (Rajavel et al., 2022).

Benefiting from recent advances in Large Language Models (LLMs) (Achiam et al., 2023; OpenAI, 2024a) and Multimodal Large Language Models (MLLMs) (Team et al., 2023; OpenAI, 2024b), VideoQA has made rapid progress. Strong MLLMs equipped with cross-modal attention, temporal modeling, and instruction-following abilities have substantially improved accuracy on standard benchmarks. Nevertheless, significant challenges remain, rooted in the very structure of our training data. Conventional datasets (Jang et al., 2017; Wu et al., 2021; Xiao et al., 2021) are composed of discrete question-answer pairs that, while factually correct, present video content as a series of fragmented, isolated facts. This format omits the rich web of inter-dependencies, such as the causal, temporal, and social links, that connect these facts into a coherent event. To highlight this fundamental limitation, Table 1 presents a typical set of human-annotated questions for a single video.

Individually, each QA pair in Table 1 provides a useful, atomic piece of information. However, their true value lies in the semantic links that are entirely ignored by conventional training paradigms. For instance, understanding why the people are resting (Q3, Q5) is contingent on knowing they are on snowmobiles (Q1). Inferring their relationship as 'friends' (Q6) is not a direct visual fact but an inference supported by the playful 'posing' interaction (Q4). Current models (Ko et al., 2023; Liang et al., 2024), trained on this data, are tasked with learning from a "bag-of-facts," forcing them to rely on shallow correlations rather than deep, structural understanding. This not only limits

generalization but is a primary cause of model hallucination when complex reasoning is required. The critical research gap, therefore, is not just the scarcity of data, but the absence of a supervision signal that represents this underlying event structure.

To address this fundamental challenge, we propose a framework that introduces two novel forms of supervision by transforming the fragmented QA pairs already present in existing datasets. Our first strategy, **Question-based Paraphrasing (QBP)**, addresses the need for structured understanding. It leverages the rich interrogative diversity (what, how, why) inherent in human-annotated questions to reverse-engineer a video's underlying event structure. Instead of treating them as a bag of isolated facts, QBP compels a LLM to synthesize these descriptive, procedural, and causal inquiries into a single, logic-infused narrative. This process transforms fragmented seeds of human curiosity into a holistic, narrative-level supervision signal. However, a global narrative alone cannot guarantee visual grounding. To this end, our second strategy, **Question-based Captioning (QBC)**, provides instance-level grounding. It generates fine-grained, question-conditioned captions that serve as visual rationales, forcing the model to anchor its reasoning in specific, relevant visual evidence. Together, QBP and QBC provide two orthogonal yet synergistic forms of supervision: one that builds a coherent narrative fabric, and another that ties each thread of that fabric to a concrete visual detail.

Extensive experiments validate the effectiveness of our approach. On two widely used benchmarks, NExT-QA and STAR (Xiao et al., 2021; Wu et al., 2021), our QBP+QBC strategies consistently improve performance across different model backbones. For example, with a Qwen2.5-VL-3B (Bai et al., 2025) backbone, accuracy on STAR improves from 67.6% to 72.5%, a gain of nearly +5 points. Larger backbones like Qwen2.5-VL-7B and MiMo-VL-SFT (Team et al., 2025) also benefit, with our QBP+QBC supervision pushing a 7B model to a new state-of-the-art of 80.8% on NExT-QA. Beyond raw accuracy, our analyses reveal significant secondary benefits: QBP's narrative supervision accelerates model convergence by more than 2.5 times, while both strategies substantially improve cross-dataset generalization, demonstrating enhanced robustness.

In summary, our contributions are as follows: (i) We propose a new supervision paradigm for VideoQA that moves beyond isolated facts, introducing two complementary synthesis strategies (QBP and QBC) to generate narrative-level and instance-level supervision. (ii) We demonstrate through large-scale experiments that our framework significantly improves both in-domain accuracy and cross-dataset generalization, achieving new state-of-the-art results on multiple challenging benchmarks. (iii) We provide a comprehensive analysis of the distinct benefits of our methods, showing that QBP accelerates model convergence by over 2.5x while both strategies enhance generalization, underscoring the efficiency and robustness of our approach.

## 2  RELATED WORK

**Video Question Answering: From Architectures to Data Bottlenecks.**  VideoQA is a challenging multimodal task requiring complex spatio-temporal reasoning. Early progress was largely driven by architectural innovations, from spatio-temporal attention mechanisms (Xu et al., 2017; Jang et al., 2017) and graph-based models (Xiao et al., 2022) to large-scale pre-trained transformers (Yang et al., 2020; Wang et al., 2022). While these models have become increasingly sophisticated, their performance is fundamentally bottlenecked by the available training data (Zhang et al., 2023; Li et al., 2023). Manually annotating large-scale, diverse, and unbiased datasets that cover complex reasoning scenarios is prohibitively expensive. Consequently, the field's focus is gradually shifting from purely architectural improvements to data-centric approaches (Liang et al., 2025) that can enhance the quality and form of the supervision signal itself.

**Data Synthesis for Video Understanding.**  Early approaches in VideoQA relied on rule-based templates (Grunde-McLaughlin et al., 2021; Wu et al., 2021) or simple question generation (Falcon et al., 2020), but these methods often produce syntactically simple and semantically repetitive data. The advent of powerful generative models has enabled more sophisticated synthesis. MLLMs like Video-LLaMA (Zhang et al., 2023) can generate descriptive video captions, while recent work such as LLaVA-Video (Zhang et al., 2024) and ShareGPT4V (Chen et al., 2024) has prompted LLMs like GPT-4 (Achiam et al., 2023) to generate a variety of video-centric textual data.

However, despite the improved quality, the dominant paradigm remains the generation of more isolated data points—be it captions or individual QA pairs. This approach enriches the dataset in

Table 1: An example of fragmented yet semantically linked QA pairs for a single video from NExT-QA. While each pair provides an isolated fact, their inter-dependencies (rightmost column) reveal a richer event structure. Conventional training paradigms ignore these crucial links, forcing models to learn from a "bag-of-facts" and hindering deep reasoning.

| Question & Answer | Question Type | Semantic Links & Implied Context |
|---|---|---|
| Q1: How are the people transported on snow? (snowmobile) | Transportation | Context for understanding the setting and actions in Q3, Q5. |
| Q2: What is the weather like? (cold) | Scene / Weather | Provides general atmospheric context for the entire scene. |
| Q3: Why is the person in red sitting on a snowmobile? (resting) | Action Reasoning | Links the action ('sitting') to a purpose ('resting'), dependent on Q1, Q5. |
| Q4: How does the man in black react to the camera? (poses) | Interaction | Implies a social relationship ('friends', Q6) and connects to camera actions (Q7, Q8). |
| Q5: Why are the snowmobiles parked? (resting) | Causal Reasoning | The overarching reason for the scene's static nature, connects to Q1, Q3. |
| Q6: What is the relationship between the people? (friends) | Social Relation | Inferred from playful interactions like 'posing' (Q4) and 'taking photos' (Q7, Q8). |
| Q7: Why is the man in blue holding a camera? (to take a photo) | Action Purpose | Explains the core interaction, directly linked to the reaction in Q4 and action in Q8. |
| Q8: What does the man in red do? (takes a photo) | Specific Action | A key interaction that supports the inference of 'friends' (Q6) and explains the 'posing' (Q4). |

volume but fails to address the core problem we identify in our introduction: the structural fragmentation of supervision. These methods do not provide the connective tissue that links discrete facts into a coherent event structure, which is essential for deep reasoning.

**Our Contribution in Context.** Our work is situated within this trend of LLM-based data synthesis but makes a distinct and complementary contribution. Instead of generating *more* fragmented data, we focus on creating *new forms* of structured supervision. We are the first to propose a dual-pronged framework that explicitly addresses the structural deficit. Our Question-based Paraphrasing introduces a novel narrative-level supervision signal, designed to reconstruct the video's event structure from existing queries. Concurrently, our Question-based Captioning provides rationale-level supervision, forcing a tight, evidence-based alignment between a specific query and its visual proof. By synthesizing these two synergistic forms of supervision, our work directly tackles the limitations of the "bag-of-facts" paradigm that characterizes prior work.

## 3 METHOD

Our work introduces a novel framework for synthesizing high-quality training data to improve VideoQA models. Instead of simply augmenting existing datasets, we propose a method to transform the sparse, fragmented supervision inherent in human-annotated QA pairs into dense, multi-level training signals. We develop two complementary synthesis techniques: Question-based Paraphrasing (QBP), which generates holistic, narrative-level supervision; and Question-based Captioning (QBC), which provides fine-grained, instance-level grounding. The overall pipeline of our method is illustrated in Figure 1.

### 3.1 PROBLEM FORMULATION

Formally, let the source data be a collection of videos $\mathcal{V} = \{v_i\}_{i=1}^N$. Each video $v_i$ is associated with a set of $K_i$ human-annotated question-answer pairs, which we denote as a question group $\mathcal{G}_i = \{(Q_{i,k}, A_{i,k})\}_{k=1}^{K_i}$. A video $v_i$ is represented as a sequence of $T$ uniformly sampled frames:

$$v_i = \{f_{i,1}, f_{i,2}, \ldots, f_{i,T}\}, \quad f_{i,t} \in \mathbb{R}^{H \times W \times 3}.$$

Our first step is to leverage the question groups $\{\mathcal{G}_i\}_{i=1}^N$ to synthesize two new datasets derived from the videos in $\mathcal{V}$:

- A narrative-level dataset, $\mathcal{D}^{\text{QBP}} = \{(v_i, \tilde{d}_i^{\text{narrative}})\}_{i=1}^N$, generated via our QBP strategy.
- A rationale-level dataset, $\mathcal{D}^{\text{QBC}} = \bigcup_{i,k} \{(v_i, \tilde{d}_{i,k}^{\text{rationale}})\}$, generated via our QBC strategy.

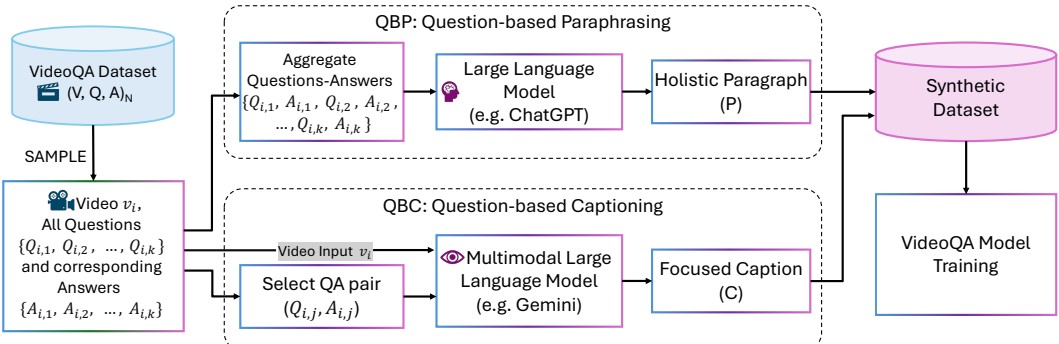

Figure 1: An overview of our framework for transforming fragmented QA pairs into structured supervision. Question-based Paraphrasing (QBP) synthesizes multiple QA pairs into a holistic narrative for global context, while Question-based Captioning (QBC) generates a visual rationale from a single QA pair to provide fine-grained, evidence-based grounding.

Crucially, our training paradigm for a model $\mathcal{M}$ does not use the original, fragmented QA pairs for supervision. Instead, our objective is to train $\mathcal{M}$ exclusively on the union of our synthesized datasets:

$$\mathcal{D}_{\text{train}} = \mathcal{D}^{\text{QBP}} \cup \mathcal{D}^{\text{QBC}}.$$

We aim to demonstrate that training on $\mathcal{D}_{\text{train}}$ yields superior performance in terms of reasoning, generalization, and grounding compared to models trained on the standard QA dataset format.

### 3.2 QUESTION-BASED PARAPHRASING (QBP): BUILDING GLOBAL NARRATIVES

A key limitation of the standard VideoQA training paradigm is its reliance on isolated question-answer pairs as supervision units. This fragmentation ignores the rich semantic dependencies that often exist between questions associated with the same video. As illustrated in Table 1, questions may be temporally, causally, or logically linked (e.g., Q1, Q3, and Q5 all concern the resting state, while Q7 and Q8 both hinge on camera-related actions). Ignoring these relations prevents models from forming a unified representation of the video's event structure.

**Conceptual Framework.** To overcome this fragmentation, we introduce Question-based Paraphrasing (QBP), a strategy designed to reconstruct the underlying event structure from these isolated annotations. Our key insight is that the set of human-annotated questions for a video, $\mathcal{G}_i$, is not a random collection of facts, but a rich sample of *interrogative diversity*. These questions probe the video's content at multiple semantic levels: 'what' questions establish static entities, 'how' questions trace dynamic processes, and 'why' questions uncover causal relationships.

QBP frames the data synthesis task as a *reasoning integration* problem. It compels an LLM to move beyond answering individual questions and instead synthesize these descriptive, procedural, and causal inquiries into a single, logic-infused narrative. This process transforms the fragmented "bag-of-facts" represented by $\mathcal{G}_i$ into a holistic, narrative-level supervision signal, $\tilde{d}_i^{\text{narrative}}$. By training on these narratives, the model is exposed to the connective tissue of the event, encouraging a shift from simple fact retrieval to structured event comprehension.

**Formalization.** Formally, given the question group $\mathcal{G}_i$ for a video $v_i$, we employ a LLM, denoted as $\Phi_{\text{QBP}}$, to generate a single narrative description $\tilde{d}_i^{\text{narrative}}$:

$$\tilde{d}_i^{\text{narrative}} = \Phi_{\text{QBP}}(\mathcal{G}_i).$$

Here, the full question-answer pairs in $\mathcal{G}_i$ are provided as input, allowing the LLM to use the ground-truth answers as factual cornerstones for its narrative reconstruction. The prompt for $\Phi_{\text{QBP}}$ explicitly instructs the model to integrate information across all QA pairs in $\mathcal{G}_i$ into a coherent, fluent paragraph, capturing latent dependencies. This process is applied to all videos in the source collection to construct the full narrative-level dataset:

$$\mathcal{D}^{\text{QBP}} = \{(v_i, \tilde{d}_i^{\text{narrative}})\}_{i=1}^{N}.$$

## 3.3 Question-based Captioning (QBC): Enhancing Visual Grounding

While QBP provides models with a global narrative context, a persistent challenge in VideoQA is *visual grounding*: ensuring answers are derived from tangible visual evidence rather than dataset biases or spurious correlations. Models often fail at fine-grained spatio-temporal localization, particularly for complex "why" or "how" questions. For example, given the question "Why did the person drop the ball?", a generic caption like "A person is playing with a ball" offers little explanatory power. In contrast, a targeted *visual rationale* such as "The person's hand slips as they try to catch the ball, causing it to fall" directly links the reasoning to an observable, causal event.

**Conceptual Framework.** To instill this level of grounding, we propose Question-based Captioning. This strategy generates fine-grained visual rationales conditioned on individual question-answer pairs. The question focuses the general topic, while the ground-truth answer provides a specific anchor for correctness. This prompts a Multimodal Large Language Model to identify and describe the precise visual evidence that *justifies* the given answer. This process creates a strong alignment between a query, its correct answer, and its visual proof, forcing the downstream model to learn not just *what* the answer is, but *why* it is correct based on the video.

**Formalization.** For each video $v_i$ and each of its associated question-answer pairs $(Q_{i,k}, A_{i,k})$ from the question group $\mathcal{G}_i$, we synthesize a targeted visual rationale $\tilde{d}_{i,k}^{\text{rationale}}$. This is generated by a Multimodal LLM, denoted as $\Phi_{\text{QBC}}$, which takes the video, the question, and the answer as input:

$$\tilde{d}_{i,k}^{\text{rationale}} = \Phi_{\text{QBC}}(v_i, Q_{i,k}, A_{i,k}).$$

Here, explicitly providing the ground-truth answer $A_{i,k}$ is a crucial design choice. It constrains the generation task, ensuring the correctness and relevance of the output. Instead of open-endedly describing the scene, the MLLM is instructed to find and articulate the specific visual evidence that supports the given correct answer. The prompt is carefully designed to forbid the model from merely repeating the answer, forcing it to generate a descriptive proof. This synthesis is performed for all question-answer pairs in the original dataset to construct the rationale-level dataset:

$$\mathcal{D}^{\text{QBC}} = \bigcup_{i=1}^{N} \bigcup_{k=1}^{K_i} \{(v_i, \tilde{d}_{i,k}^{\text{rationale}})\}.$$

This dataset consists of '(video, text)' pairs, structurally identical to $\mathcal{D}^{\text{QBP}}$, where each text serves as a grounded explanation for an implicit question-answer pair.

**Complementary Nature.** Conceptually, QBC complements QBP. Whereas QBP focuses on constructing holistic narratives that capture global dependencies across multiple questions, QBC operates at a fine-grained level, enforcing a tight alignment between an individual query-answer pair and its supporting visual evidence. Together, they provide two orthogonal yet synergistic forms of synthetic supervision, namely global narrative coherence and local visual grounding, which constitute our final training set $\mathcal{D}_{\text{train}} = \mathcal{D}^{\text{QBP}} \cup \mathcal{D}^{\text{QBC}}$.

## 4 Experiments

In this section, we conduct comprehensive experiments to empirically validate our proposed data synthesis framework. Our evaluation is structured to answer several key questions regarding its effectiveness, properties, and the quality of its outputs. We first describe our experimental setup and then present a human evaluation to assess the quality and factual fidelity of the generated supervision signals, including an analysis of the seed datasets and the statistical properties of our synthetic data. Finally, we evaluate model performance and analyze the contributions of different components in our framework.

**Training.** All models are fine-tuned exclusively with a next-token prediction objective. For fair comparison, hyperparameters are kept consistent across all experimental settings. We use the AdamW optimizer with a learning rate of 1e-6 and train for 1-2 epochs. For video processing, we uniformly sample 16 frames. See more details in Appendix B.

**Evaluation Metrics.** Our primary metric is *Accuracy*, calculated via exact match with ground-truth answers. To assess generalization, we perform cross-dataset evaluation, where a model is trained on one dataset (e.g., NExT-QA) and tested on another unseen dataset (e.g., STAR).

## 4.1 SYNTHETIC DATA QUALITY ASSESSMENT

**Data Analysis.** Our data synthesis process begins with three widely-used VideoQA benchmarks as seeds: NExT-QA, STAR, and DiDeMo (Xiao et al., 2021; Wu et al., 2021; Anne Hendricks et al., 2017). As shown in Table 2, these datasets exhibit notably different annotation densities. NExT-QA and STAR provide relatively dense supervision, with an average of 9 and 15 QA pairs per video, respectively. In contrast, DiDeMo is much sparser, with only 3.5 QA pairs per video. This disparity is further visualized in Figure 2. These statistics underscore the complementary nature of our proposed methods. The high density of questions in datasets like STAR provides a rich source for QBP to consolidate into coherent narratives. Conversely, the sparsity of datasets like DiDeMo highlights the need for QBC to expand annotation coverage with fine-grained, grounded descriptions.

Table 2: Statistics of the datasets we used.

|          | #video | #QA(Annotation) |
|----------|--------|-----------------|
| NExT-QA  | 3.8k   | 34k             |
| STAR     | 3k     | 45k             |
| DiDeMo   | 2k     | 7k              |

Figure 2: Distribution of the number of QA pairs per video across datasets. NExT-QA and STAR include a wide range of annotations per video, with some clips having more than 20 questions, while DiDeMo remains consistently sparse.

Figure 3: Length distributions of textual supervision before and after synthesis. Raw QA pairs are short and fragmented. QBP generates moderately longer narratives with high semantic density, while QBC produces the longest, fine-grained captions.

Next, we analyze the textual properties of our synthesized data. Figure 3 compares the length distributions. The original answers are predominantly short and fragmented. In contrast, QBP narratives are moderately longer and exhibit high semantic density, while QBC rationales produce the longest and most detailed descriptions. This analysis confirms that our framework successfully transforms sparse annotations into two distinct and complementary forms of supervision: one focused on semantic density (QBP) and the other on descriptive richness (QBC).

**Quantitative Human Evaluation.** While our synthesis process is seeded with human-annotated QA pairs, the LLM or MLLM generator could potentially introduce errors. To rigorously evaluate this, we conduct a human evaluation study on the quality of our generated data. We randomly sample 100 QBP narratives and 100 QBC rationales and ask three human evaluators to rate them on a 1-5 Likert scale across several key dimensions. Instructions for human evaluators are in the App. D.1.

As shown in Table 3, our synthesized data achieves consistently high scores. Both QBP and QBC demonstrate strong *Factual Consistency* (4.21 and 4.35, respectively), confirming that the LLM and MLLM generally preserve the ground-truth information from the source QA pairs. QBP narratives are rated favorably for *Logical Coherence* (4.25), while QBC rationales receive a high score for *Visual Grounding* (4.38). The low standard deviation across most metrics, particularly

Table 3: Human evaluation of synthetic data quality on a 1-5 scale. The scores are consistently high, confirming the overall quality and fidelity of the synthesized data.

| Evaluation Dimension | QBP           | QBC           |
|----------------------|---------------|---------------|
| Factual Consistency  | $4.21 \pm 0.55$ | $4.35 \pm 0.48$ |
| Logical Coherence    | $4.25 \pm 0.61$ | -             |
| Visual Grounding     | -             | $4.38 \pm 0.52$ |
| Fluency              | $4.88 \pm 0.21$ | $4.91 \pm 0.19$ |

*Fluency*, indicates strong agreement among evaluators on the high quality of the generated text. These results confirm that our synthesis process produces reliable supervision signals suitable for model training.

**Qualitative Error Analysis.** We perform a qualitative analysis of failure cases to better understand the limitations of our approach. We find that severe errors, such as hallucinating non-existent events, are extremely rare. The more common, though still infrequent, failure modes are subtle and method-specific. For QBP, the primary challenge lies in logical cohesion. We observe occasional errors in temporal ordering, where the LLM incorrectly sequences two closely related actions. This

Table 4: Model comparison on NExT-QA and STAR. All scores are reported in Accuracy (%).

| Model | LLM Arch. | NExT-QA | STAR |
|---|---|---|---|
| *Fine-tuned on Raw QA pairs* | | | |
| InternVideo (Wang et al., 2022) | - | 63.2 | 58.7 |
| LSTP (Wang et al., 2024) | FlanT5 3B | 72.1 | - |
| VidF4 (Liang et al., 2024) | FlanT5 3B | 74.1 | 68.1 |
| LLaMA-VQA (Ko et al., 2023) | LLaMA 7B | 72.0 | 65.4 |
| MotionEpic (Fei et al., 2024) | Vicuna 7B | 76.0 | 71.0 |
| Vamos (Wang et al., 2023) | LLaMA2 7B | 75.0 | - |
| LLaVA-OV(Li et al., 2024) | Qwen2 7B | 77.5 | 66.2 |
| *Combined with on QBP+QBC (ours)* | | | |
| Qwen2.5-VL (Bai et al., 2025) | Qwen2.5 3B | 74.3 | 67.5 |
| w. QBP+QBC (ours) | Qwen2.5-3B | 76.8 (+2.5) | 72.5 (+5.0) |
| MiMo-VL-SFT (Team et al., 2025) | MiMo 7B | 75.3 | 52.0 |
| w. QBP+QBC (ours) | MiMo 7B | 77.0 (+1.7) | 56.2 (+4.2) |
| Qwen2.5-VL (Bai et al., 2025) | Qwen2.5 7B | 76.2 | 70.6 |
| w. QBP+QBC (ours) | Qwen2.5 7B | 80.8 (+4.6) | 73.3 (+2.7) |

issue is sometimes exacerbated by imprecise temporal boundary annotations in the source QA pairs themselves, which provide ambiguous cues. In rarer cases, we note entity confusion, where a single person is described with conflicting pronouns as if they were two separate individuals. For QBC, the most notable failure mode is a form of "justified fabrication." Since the MLLM is provided with the correct answer, it sometimes invents plausible-sounding visual details to rationalize the answer, especially when the actual visual evidence is subtle or ambiguous. Detailed examples are provided in the Appendix C.2.

## 4.2 MAIN PERFORMANCE EVALUATION

We evaluate our data synthesis framework by comparing it against previously published state-of-the-art (SOTA) models (e.g., Vamos (Wang et al., 2023), MotionEpic (Fei et al., 2024)), which are fine-tuned on the original QA training sets of each benchmark. Further details on these baselines are provided in Appendix B. For our evaluation, we select two representative MLLMs, Qwen2.5-VL (Bai et al., 2025) and MiMo-VL (Team et al., 2025), chosen for their strong general-purpose reasoning ability. Table 4 summarizes the results on NExT-QA and STAR.

The results are clear and consistent: across all backbones and model scales, training exclusively on our synthesized data provides significant performance improvements. For example, when applied to the Qwen2.5-VL-3B, our method boosts accuracy on NExT-QA from 74.3% to 76.8% (+2.5) and delivers a remarkable +5.0 point gain on STAR, increasing accuracy from 67.5% to 72.5%. This trend holds for larger 7B models as well; notably, our method pushes the Qwen2.5-VL-7B model to a new SOTA of 80.8% on NExT-QA. Importantly, even when built upon strong backbones, our synthesized supervision consistently outperforms models fine-tuned on the raw QA pairs, underscoring its effectiveness and general applicability.

This consistent improvement validates our core hypothesis: transforming the supervision format from a "bag-of-facts" into structured, multi-level signals is a more effective way to train VideoQA models. The narrative-level context from QBP enables models to better understand temporal and causal event structures, while the instance-level grounding from QBC forces a tighter alignment between reasoning and specific visual evidence. This richer supervision allows models to move beyond shallow pattern matching and develop a deeper, more robust comprehension of video content, leading to higher accuracy on complex reasoning tasks.

## 4.3 IN-DEPTH ANALYSIS OF QUESTION-BASED PARAPHRASING (QBP)

In this section, we conduct a detailed analysis to understand the properties of QBP. We aim to answer two key questions: (1) How does fine-tuning on QBP-synthesized narratives compare to fine-tuning

Table 5: Comparison of fine-tuning on raw QA pairs vs. our QBP-synthesized narratives. QBP effectively improves in-domain performance while also enhancing cross-domain generalization, mitigating the overfitting seen with raw data.

Table 6: Effect of using different and combined seed datasets for QBP synthesis. Performance is evaluated on NExT-QA and STAR. Combining diverse seeds yields the best generalization.

| QBP Seed Data Source(s) | Test on NExT-QA | Test on STAR |
|---|---|---|
| *Qwen2.5-VL-3B (No fine-tuning)* | *74.3* | *67.6* |
| NExT-QA only | 76.0 | 69.8 |
| DiDeMo only | 76.0 | 69.1 |
| STAR only | 75.5 | 69.9 |
| NExT-QA + DiDeMo | 76.3 | 69.4 |
| NExT-QA + STAR | 76.2 | **70.9** |
| NExT-QA + DiDeMo + STAR | **76.5** | 70.8 |

| Training Data | Test on NExT-QA | Test on STAR |
|---|---|---|
| *Qwen2.5-VL-3B* | *74.3* | *67.6* |
| NExT-QA (raw) | 76.2 (+1.9) | 66.5 (-1.1) |
| **QBP from NExT-QA** | 76.0 (+1.7) | 69.8 (+2.2) |
| STAR (raw) | 73.1 (-1.2) | 70.2 ((+2.6) |
| **QBP from STAR** | 75.5 (+1.2) | 69.9 ((+2.3) |

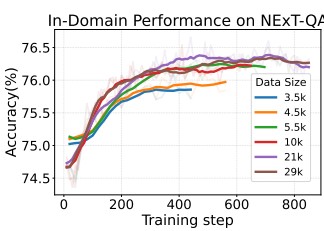

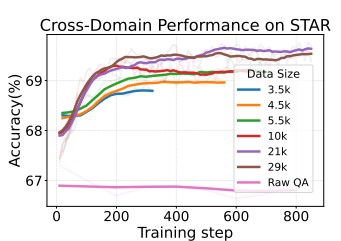

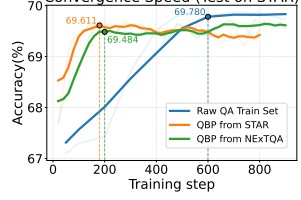

Figure 5: Convergence with raw QA vs. QBP. QBP enables faster convergence compared to raw QA training, showing the efficiency of holistic descriptions.

Figure 4: Effect of QBC scale. Larger amounts of synthesized QBC data improve accuracy and convergence on both NExT-QA and STAR, with clear gains in cross-dataset generalization.

on raw QA pairs, particularly concerning cross-dataset generalization? (2) How does the choice of seed dataset for synthesis affect QBP's performance?

**QBP Mitigates Overfitting from Raw Data.** A common risk in fine-tuning is overfitting to the source dataset's specific patterns and biases. To investigate whether QBP can mitigate this issue, we compare models fine-tuned on raw QA pairs from a single source against models fine-tuned on QBP narratives synthesized from that same source.

Table 5 reveals a critical trend. As expected, fine-tuning the backbone on the raw NExT-QA training set improves its in-domain performance significantly (+1.9%), but this comes at the cost of degraded performance on the unseen STAR dataset (-1.1%), a clear sign of overfitting. Conversely, training on QBP narratives synthesized from NExT-QA not only boosts in-domain accuracy but also enhances cross-dataset generalization to STAR (+2.2%). The same pattern holds when using STAR as the source dataset. This directly validates that the narrative supervision from QBP provides a more generalizable signal than the original, fragmented QA pairs.

**Effect of Diverse Seeds for QBP Synthesis.** Next, we explore how leveraging a diverse mix of seed datasets for QBP synthesis impacts performance. We generate QBP narratives using various combinations of NExT-QA, STAR, and DiDeMo as source material. As shown in Table 6, combining seeds from multiple datasets yields the most significant gains, particularly for cross-domain generalization. While narratives from a single source already provide benefits, synthesizing from a mix of NExT-QA and STAR pushes the STAR accuracy to a high of 70.9% (+3.3% over the backbone). Incorporating all three diverse sources (NExT-QA, STAR, DiDeMo) achieves the best overall balance, reaching 76.5% on NExT-QA and 70.8% on STAR. This confirms that QBP is most effective when it can draw upon a wide range of question styles and content, allowing it to generate a richer and more robust narrative supervision signal that transcends the biases of any single dataset.

**Accelerated Convergence with QBP's Narrative Supervision.** A striking finding of our study concerns the training efficiency of QBP. As shown in Figure 5, models trained on QBP-synthesized narratives converge dramatically faster than those trained on the original, fragmented QA pairs. For

instance, the NExT-QA training set consists of approximately 30k individual QA pairs, which our QBP process condenses into about 3k holistic paragraphs. Despite this ten-fold reduction in the number of training instances, the model trained on QBP data reaches its performance plateau within approximately 220 steps. In stark contrast, the model trained on the raw QA set requires around 600 steps to reach a similar performance level. This demonstrates that our QBP-based supervision accelerates convergence by more than 2.5x compared to the standard QA training paradigm.

This accelerated convergence may seem counterintuitive, given that QBP narratives are textually longer than individual QA pairs. We hypothesize this is because the narrative supervision acts as a far more semantically dense and informative training signal. Each paragraph synthesizes multiple related inquiries ('what', 'how', 'why') and their dependencies into a unified context that reflects the video's underlying event structure (see Table 1). This provides the model with richer, more structured reasoning cues in a single optimization step, effectively reducing the redundancy inherent in processing numerous overlapping, low-level QA pairs.

From a practical standpoint, this highlights a significant efficiency advantage of QBP. In resource-constrained settings, the ability to reach a high-performance state with fewer training steps makes QBP-based supervision a particularly appealing and cost-effective strategy.

### 4.4 Analysis of QBC: Data Scaling and Generalization

Having analyzed QBP's properties with respect to seed diversity, we now turn to QBC and investigate its effectiveness as a function of data scale. We synthesize varying amounts of QBC rationales from the NExT-QA training set (from 3.5k up to the full 29k samples) and fine-tune the Qwen2.5-VL-3B backbone on each subset. Performance is monitored on both the in-domain (NExT-QA) and cross-domain (STAR) test sets.

The results, plotted in Figure 4, show a clear and positive correlation between the volume of synthetic data and model performance. (1) **In-domain Performance (Fig. 4a):** On NExT-QA, accuracy steadily improves as more QBC rationales are added. With just 5k samples, the model already surpasses the baseline, and performance continues to climb as the dataset scales to 10k and then 29k samples. This confirms that the fine-grained, grounded supervision provided by QBC offers a strong and scalable training signal for improving in-domain reasoning. (2) **Cross-dataset Generalization (Fig. 4b):** The benefits of scaling QBC data are even more pronounced in the cross-dataset transfer setting. On STAR, performance rises from a baseline of 66.5% (*Raw QA*) to nearly 70.0% with the full 29k set, a gain of +3.5 points. This striking trend demonstrates that training on QBC's visual rationales effectively forces the model to ground its predictions in visual evidence rather than source-specific linguistic biases, thereby enhancing its ability to generalize to new, unseen domains.

In summary, this analysis validates QBC as a highly scalable form of supervision. Increasing the volume of QBC data consistently improves both in-domain accuracy and, critically, cross-dataset generalization. This highlights its role as a powerful tool for generating fine-grained, evidence-based supervision that complements the holistic, narrative context provided by QBP.

## 5 Conclusion

In this work, we present a novel data-centric paradigm for VideoQA that moves beyond the limitations of training on isolated, factual annotations. Our framework introduces two complementary synthesis strategies: Question-based Paraphrasing (QBP), which generates coherent, narrative-level supervision, and Question-based Captioning (QBC), which provides fine-grained, instance-level visual grounding. Our extensive experiments demonstrate that training models exclusively on this synthesized data establishes a new state-of-the-art on multiple challenging benchmarks. Beyond accuracy, we show that our method yields significant secondary benefits: it substantially enhances cross-dataset generalization, and the narrative supervision from QBP markedly accelerates model convergence by more than 2.5x. Our rigorous human evaluation further confirms the high factual consistency and logical coherence of the synthesized data, solidifying its reliability as a high-quality supervision signal. These results highlight the profound potential of shifting focus from model architecture to the supervision signal itself. By transforming fragmented inquiries into structured narratives and grounded rationales, we unlock significant gains in model performance, robustness, and training efficiency.

## ETHICS STATEMENT

This work builds on publicly available VideoQA datasets, which contain human-annotated QA pairs and captions. No private or sensitive data are used. Our data synthesis strategies (QBP and QBC) rely on large language models and multimodal models to generate additional supervision, but the generated content remains constrained to the semantics of the original annotations, reducing risks of misinformation or harmful outputs. Potential societal risks include over-reliance on synthetic data or propagation of biases from source models; we mitigate this by grounding synthesis in human-verified annotations and reporting transparent analyses. All experiments follow standard academic use of benchmarks and are intended solely for advancing research in multimodal reasoning.

## REPRODUCIBILITY STATEMENT

We provide detailed descriptions of our methods, datasets, and experimental settings to ensure reproducibility. Specifically, we outline the backbone architectures, frame sampling strategy, training objectives, and hyperparameters. Dataset splits follow publicly available benchmarks. Prompts used for QBP and QBC synthesis are included in the Appendix. We also report results averaged across multiple random seeds to account for variance. All resources required to reproduce our results including code, and processed data will be released upon publication.

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

## A  USE OF LARGE LANGUAGE MODELS

arge language models are used in two ways in this work. First, they support data synthesis, where QBP relies on language models to paraphrase human-annotated QA pairs into narrative form, and QBC employs multimodal models to generate query-conditioned captions. Second, they play a supportive role in writing, including proofreading, correcting grammatical errors, and improving clarity of exposition. All model outputs are carefully reviewed by the authors, and responsibility for the final content rests entirely with the authors.

## B  EXPERIMENTAL DETAILS

**Training details**  We finetune model using the SFTTrainer from TRL [1] and DeepSpeed [2] during training in NVIDIA H800 (80GB) GPU $\times$ 2. We use AdamW with a cosine learning rate scheduler, whose max learning rate is 1e-6, and a batch size of 8. We train our model within 1-2 epochs. Our training code will be later open sourced.

---

[1] https://huggingface.co/docs/trl/v0.22.1/en/sft_trainer
[2] https://github.com/deepspeedai/DeepSpeed

**Baselines.** We evaluate our data synthesis framework by comparing it against previously SOTA models, such as InternVideo (Wang et al., 2022), LLaMA-VQA (Ko et al., 2023), LSTP (Wang et al., 2024), VidF4 (Liang et al., 2024), Vamos (Wang et al., 2023), MotionEpic (Fei et al., 2024) and LLaVA-OV Li et al. (2024). Among these models, LLaMA-VQA, Vamos, and MotionEpic use 7B-parameter LLM as part of the model.

**LSTP** adopts the BLIP-2 architecture and uses optical flow for frame selection, followed by using LLM to generate answers. Similarly, VidF4 (Liang et al., 2024) update its model by training on raw QA pairs after extracting key frames from videos.

**LLaMA-VQA** is built based on LLaMA-7B (Touvron et al., 2023), enabling the model to understand the complex relationships between videos, questions, and answers by constructing multiple auxiliary tasks.

**MotionEpic** breaks down the raw intricate video reasoning problem into a chain of simpler sub-problems and solves them one by one sequentially.

**Vamos** (Wang et al., 2023) generalizes the concept bottleneck model to work with tokens and non-linear models, which uses hard attention to select a small subset of tokens from the free-form text as inputs to the LLM reasoner.

**LLaVA-OV** (Li et al., 2024) builds upon LLaVA by constructing synthetic data to further enhance the base model. Liang et al. (2025) fine-tune the model on raw QA pairs for VideoQA tasks. However, the official paper also shows that current MLLM backbones may overlap with common benchmarks; see the original work for details.

## C PROMPTS AND EXAMPLES FOR QBP AND QBC

### C.1 PROMPTS AND EXAMPLES FOR DATA SYNTHESIS

Here, we provide the detailed prompts and concrete examples used for our data synthesis strategies.

#### C.1.1 QUESTION-BASED CAPTIONING (QBC)

The QBC prompt instructs the MLLM to generate a visual rationale—a caption that describes the visual evidence supporting a given answer, without explicitly stating the answer itself.

---

**Prompt for QBC**

```
 Given a video, a question, and its answer, generate a natural
language caption that highlights the visual content most
relevant to justifying the answer.

The caption should be a descriptive proof grounded in visual
evidence, NOT a direct restatement of the answer.
```

---

#### C.1.2 QUESTION-BASED PARAPHRASING (QBP)

The QBP prompt is designed to instruct the LLM (DeepSeek, GPT-4o) to act as a reasoning integrator, synthesizing a holistic narrative from a collection of fragmented QA pairs.

**Prompt for QBP**

```
Transform the following QA pairs into a single, logically
coherent paragraph in present tense.  Follow these rules
strictly:

1.  **Content Requirements**:
- Use ONLY information from the provided QA pairs.  Do not
invent new facts.
- If questions imply clear chronological order (e.g.,
"before"/"after"), preserve it.
- If no temporal relationship exists (e.g., between weather
and object questions), present facts in neutral order without
implying sequence (avoid "first"/"then"/"while").
- Group related facts by theme (e.g., environment → actions →
social interactions).
2.  **Prohibitions**:
- Never assume unstated temporal/causal relationships.
- Avoid speculative terms like "probably", "might", or "seems".
- Do not use filler phrases like "the questions ask about...".

**QA Pairs**:
{QAGroup}

**Output**:
```

**Example.**   To illustrate QBP's ability to reconstruct a complex event structure, we use the same set of questions previously shown in Table 1 of the introduction.

**Input QA Pairs for QBP (from Table 1)**

- Q1: How are the people transported on snow? (snowmobile)
- Q2: What is the weather like? (cold)
- Q3: Why is the person in red sitting on a snowmobile? (resting)
- Q4: How does the man in black react to the camera? (poses)
- Q5: Why have the snowmobiles parked? (resting)
- Q6: What is the relationship between the people? (friends)
- Q7: Why is the man in blue holding a camera? (to take a photo)
- Q8: What does the man in red do? (takes a photo)

**Synthesized QBP Narrative**

> In cold weather conditions, a group of friends is transported on a snowmobile across the snow. They come to a halt and park the snowmobile to rest. The man wearing red, after placing his helmet on the motorbike, takes a photo. Meanwhile, the person dressed in red sits on the snowmobile, resting alongside the group. The man in black strikes a pose when the man in red raises his camera at the end of the video. In the group, there is also a man wearing blue who holds a camera to take a photo.

This example demonstrates how QBP synthesizes descriptive ('cold', 'snowmobile'), causal ('resting'), and social ('friends', 'poses') information from the fragmented QA pairs into a single, logic-infused narrative.

Figure 6: An example of synthesized data generated by our framework. The left shows the raw QA pairs from NExT-QA, while the right presents the corresponding outputs: a narrative produced by the QBP pipeline and rationales generated by the QBC pipeline.

## C.2 EXAMPLES

To offer a more concrete understanding of our approach, this section showcases several examples generated by our proposed QBP and QBC methods. As illustrated in Figures 6 and 7, these examples highlight the practical output and effectiveness of our techniques. As noted in Section 4.1, while the proportion of imperfections remains small, occasional issues are unavoidable. For instance, in Figure 7, semantic overlap within the question group (e.g., Q2 and Q10) causes the QBP-generated narrative to include redundant concluding sentences.

## D  ASSESSING THE QUALITY OF SYNTHETIC DATA

### D.1  HUMAN EVALUATION

To quantitatively assess the quality of our synthesized data, we design and conduct a human evaluation study. We refer to QBP as Task A (Narrative Evaluation) and QBC as Task B (Rationale Evaluation). For each task, evaluators are asked to rate each generated text on a 1–5 scale across several quality dimensions, guided by the detailed descriptions provided. A score of 5 indicates the highest quality, while 1 indicates the lowest.

---

**1. Factual Consistency**

**Guiding Question:** Does the generated text contradict any of the facts provided in the source information (the QA pairs for Task A; the video and correct answer for Task B)?

- **5 (Excellent):** The text is perfectly consistent with all source facts.
- **3 (Moderate):** The text contains minor inaccuracies or makes claims that are plausible but not directly supported by the source.
- **1 (Poor):** The text directly contradicts a key fact from the source (e.g., says "the person is running" when the answer is "walking").

1100/4180731134.mp4 (NExT-QA)

timeline

Generated by QBP pipeline:

In the video, two dogs, one black and one white, are **playing** in the garden. The man swings his arm at the beginning, acting like he is **playing fetch**, prompting the white dog to **follow the direction of the swing**. After swinging his arms, the man stands close to the dogs. In the middle of the video, he bends down, making a hand gesture as if to throw something. The dogs respond to these actions by looking towards the **direction of the man's swing** and running around. As the video progresses, the black dog **chases** the white dog towards the grass. Near the end, they are playfully **chasing** each other. Eventually, the white dog runs to the owner and then jumps onto a chair. By the end, the white dog also jumps onto a bench to sit and rest.

Raw QA pair group for 1100/4180731134.mp4:

Q1: why did the black dog ran after the white dog ran towards the grass near the end of the video? (playing with each other.)
Q2: "what did the white dog do after it ran to the owner at the end of the video? (jump on the chair.)
Q3: "why did the man swing his arm? (act like playing fetch.)
Q4: "why did the white dog ran towards the grass when the man swung his hand in the middle of the video? (follow the direction of man swinging.)
Q5: "how are the black and white dogs interacting with each other near the end of the video? (chasing each other.)
Q6: "where are the dogs playing? (garden.)
Q7: "what did the man do after he swung his arms in the beginning of the video? (stands close to the dogs.)     Sample this QA pair
Q8: "why did the white smaller dog jump onto the bench in the end? (sit down and rest.)
Q9: "how did the dogs reacted when the man swing his arms in the middle of the video? (look towards the direction of swing.)
Q10: "how did the white dog get on to the bench at the end of the video?(jumped up.)
Q11: "why did the man bend down in the middle of the video? (hand gesture to throw.)
Q12: "how are the dogs moving around in the video? (running.)

Generated by QBC pipeline:

The video captures a playful scene in a sunny backyard featuring a man and three dogs. The setting includes green grass, a concrete patio with hexagonal patterns, and lush green foliage along a fence line in the background. The man, wearing a blue t-shirt and khaki shorts, initially interacts with a light-colored dog (likely a Golden Retriever or similar mix) and two smaller, fluffy dogs one black and one white/tan. He throws a small toy, and the light-colored dog enthusiastically jumps and retrieves it, bringing it back to the patio. While the larger dog is fetching, the two smaller dogs engage in playful interactions with each other on both the grass and the concrete patio. The man attempts to encourage the smaller dogs to join in the fetching game, addressing them and holding the toy. When he eventually throws the toy again, both the light-colored dog and the small white dog chase after it into the grassy area, while the black fluffy dog remains on the patio. Towards the end of the video, as the play seems to wind down, all the dogs are back on the patio. The small white dog then **jumps onto a wooden bench on the patio, appearing to sit down and rest**, as it seems tired from the activity. The man stands near the bench, with the black fluffy dog by his feet, observing the scene."

Figure 7: An example of synthesized data generated by our framework. The left shows the raw QA pairs from NExT-QA, while the right presents the corresponding outputs: a narrative produced by the QBP pipeline and rationales generated by the QBC pipeline. Due to overlapping content in the question group (e.g., Q2 and Q10), the generated QBP narrative includes two similar concluding sentences, marked in blue and green for clarity, reflecting minor redundancy introduced by semantically repetitive QA pairs.

---

**2. Logical Coherence (Task A - QBP only)**

**Guiding Question:** Does the narrative describe events in a logical and coherent order? Does the story make sense?

- **5 (Excellent):** The sequence of events is clear, logical, and easy to follow. Causal and temporal relationships are sensible.

- **3 (Moderate):** The narrative is generally understandable, but the ordering of some events might be slightly awkward or ambiguous.

- **1 (Poor):** The narrative is confusing, jumbled, or illogical (e.g., describes an effect before its cause, or confuses the identities of different people).

---

**3. Visual Grounding (Task B - QBC only)**

**Guiding Question:** Does the rationale describe specific, observable evidence from the video that helps to justify the given answer?

- **5 (Excellent):** The rationale perfectly describes tangible visual details that serve as strong, direct evidence for the answer.

- **3 (Moderate):** The rationale is relevant but somewhat generic, describing the general scene rather than the specific evidence.

- **1 (Poor):** The rationale is irrelevant, describes something not visible in the video (fabrication), or simply rephrases the question without providing visual evidence.

## 4. Fluency

**Guiding Question:** Is the generated text well-written, grammatically correct, and easy for a native speaker to read?

- **5 (Excellent):** Flawless grammar and natural, fluent phrasing.
- **3 (Moderate):** Contains minor grammatical errors or awkward phrasing that do not impede understanding.
- **1 (Poor):** The text is ungrammatical, nonsensical, or very difficult to understand.

