# OpenReview forum: "Beyond Isolated Facts: Synthesizing Narrative and Grounded Supervision for VideoQA"
_ICLR.cc/2026/Conference — Submitted to ICLR 2026_

### Official Review · Reviewer_BjEU · 2025-10-22

**Soundness:** 2
**Presentation:** 3
**Contribution:** 2
**Rating:** 4
**Confidence:** 3

**Summary:**

This paper proposes to transform QA pairs into paragraph / captioning supervision, specifically via:
Question-Based Paraphrasing (QBP): This method uses a LLM to synthesize all the individual QA pairs associated with a single video into one cohesive, holistic narrative paragraph.
Question-Based Captioning (QBC): This method uses a VLM to generate a fine-grained visual rationale for each individual QA pair.
The authors train VideoQA models exclusively on this newly synthesized data (a mix of QBP narratives and QBC rationales) using a standard next-token prediction objective. Experiments on the NExT-QA and STAR benchmarks show that this method achieves new sota, as well as cross-dataset generalization and faster model convergence by over 2.5x.

**Strengths:**

- Good problem formulation and intuitive approach
- Storng results on NeXT-QA and STAR

**Weaknesses:**

- The approach is only validated on two video QA benchmarks, e.g. it is unclear if it extends well to EgoTempo, Minerva, LVBench? It is especially important since the paper claims superior cross dataset transfer.
- Lack of ablations showing the effect of only one task or how the two tasks are balanced together in the model training

**Questions:**

- How much of the performance boost is due to distilling from a potentially stronger LLM/VLM? Does the approach work when using the same model for generation and training?
- Is it correct that the model is trained uniquely with captioning objectives and inferred for video QA? Does the model also exhibit stronger captioning?

---

### Official Review · Reviewer_9MCa · 2025-10-26

**Soundness:** 2
**Presentation:** 3
**Contribution:** 2
**Rating:** 4
**Confidence:** 4

**Summary:**

This paper addresses a fundamental limitation in current Video Question Answering (VideoQA) supervision, where training typically relies on fragmented and independent QA pairs that fail to capture the underlying narrative and causal relationships among events. To mitigate this issue, the authors propose two data synthesis strategies, Question-Based Paraphrasing (QBP) and Question-Based Captioning (QBC), designed to produce richer and more contextually grounded supervision signals. Specifically, QBP merges all QA pairs for a given video into a coherent paragraph that encodes descriptive, procedural, and causal dependencies, while QBC generates question-conditioned visual rationales that explicitly ground each answer in observable video evidence. Experiments on the STAR and NExT-QA benchmarks demonstrate consistent performance improvements, achieving 72.5% accuracy on STAR with a 3B model (+4.9) and 80.8% on NExT-QA with a 7B model, along with enhanced cross-dataset generalization and approximately 2.5× faster convergence attributed to QBP-based supervision.

**Strengths:**

1. The paper presents a clear and well-motivated problem statement. It convincingly argues that conventional “bag-of-facts” supervision limits a model’s ability to capture narrative and causal dependencies within videos and effectively illustrates this issue through concrete examples of interdependent QA pairs.

2. The proposed approach is conceptually simple yet broadly generalizable. The Question-Based Paraphrasing (QBP) and Question-Based Captioning (QBC) strategies repurpose existing QA annotations into richer, narrative- and rationale-level supervision without requiring modifications to model architectures, thereby enhancing compatibility with diverse VideoQA backbones.

3. The method demonstrates strong and consistent empirical performance. The improvements observed across multiple datasets and model scales substantiate the effectiveness and robustness of the proposed data-centric supervision strategies.

**Weaknesses:**

1. The proposed synthesis pipeline heavily depends on LLMs/VLMs for generating QBP and QBC data, making it costly expensive, especially given that the original datasets already require human annotations, which are labor-intensive. Reducing this reliance on powerful models and manual labels would improve scalability and accessibility. Additionally, the paper does not explore how different LLM/VLM variants affect the quality or consistency of the synthesized QA data, which would be important for understanding robustness.

2. While the approach is claimed to be generalizable, its evaluation primarily centers on NExT-QA and STAR, with limited evidence of broader applicability. Extending the experiments to additional datasets, such as instructional, long-form, or densely annotated video corpora, would more convincingly demonstrate the generality and potential limits of the proposed framework.

3. Although the paper highlights faster convergence during fine-tuning, it lacks a thorough cost–benefit analysis of the synthesis process itself. Quantifying the computational and monetary cost of generating QBP/QBC supervision at scale (e.g., when using GPT/Gemini-series) relative to the downstream performance gains would provide a clearer understanding of the practical trade-offs involved.

**Questions:**

1. How does the number of input frames affect model performance? The reported baseline results appear notably lower than those of the original Qwen2.5-VL when using a larger frame count. It would be helpful if the authors could clarify whether the reduced frame sampling contributes to this performance gap and, if so, provide an analysis of performance trends with respect to the number of frames.

---

### Official Review · Reviewer_pZ6A · 2025-10-28

**Soundness:** 1
**Presentation:** 2
**Contribution:** 1
**Rating:** 2
**Confidence:** 3

**Summary:**

The paper proposes data augmentation techniques for Video QA datasets. Conventionally, there are multiple independent VQAs for the same visual context. This work introduces new annotations that connects each QAs semantically to build cohesive narrative, and further connecting QA with the visual context via question-conditional descriptive captions. Both are auto generated with off-the-shelf LLMs and MLLMs, respectively. When models are trained to generate these additional annotations, the models outperform baselines trained with only the independent VQAs on downstream video QA benchmarks. Further, these extended annotations also enhance generalization across datasets and accelerates convergence.

**Strengths:**

-	It is intuitive that building a cohesive narrative out of independent VQAs and using it as training target would result in a better generalization, since that aligns better with LLM's next-token prediction objective and make it more on policy due to fluency in the connective phrases.

**Weaknesses:**

-	By weaving isolated factual elements (VQAs) into a cohesive paragraph, the model effectively constructs a hypothesized interpretation of the scene depicted in the visual input. However, this approach poses several issues: (1) multiple plausible hypotheses may exist that are not determinable from the visuals alone, (2) hallucinated content can emerge during narrative construction, and (3) when using LLMs for data generation, the model may fail to produce a coherent or faithful hypothesis—a limitation frequently observed in MLLMs (e.g., [5]). None of these potential failure modes are adequately addressed in the paper, apart from a human evaluation of output quality, which itself has methodological weaknesses.
-	One of the proposed data augmentation methods, Question-Based Captioning (QBC), essentially revisits earlier works on question-conditioned captioning (e.g., [1,2]). However, the paper provides no discussion of how its approach relates to or differs from these prior studies.
-	Also, the other data augmentation method, QBP, is essentially converting a set of VQA pairs for the same visual context into a cohesive narrative. Such approach is widely studied in the literature (e.g., [3,4]), of which this paper does not talk about.
-	The choice of benchmarks are outdated; all three benchmarks tested in the paper dates back to 2021, before even the release of multimodal LLMs (e.g., GPT4V or Llava: 2023). Recent Video LLMs are tested on more recent and competitive benchmarks as MLVU, LongVideoBench, or VideoMME.
-	The performance is not impressive as the paper claims: while the paper claims SOTA performance on NextQA with 80.8% accuracy using 7B model, Llava-Video (TMLR 06/2025) already achieves 83.2% with 7B. Also, Llava-ov performance is reported as 77.5% here, while the original paper reports 79.4%.
-	The human evaluation protocol in Section 4.1 is problematic. Absolute scoring from human raters is well known to suffer from annotator bias (some raters may be systematically more lenient or strict) so such scores are meaningless without proper calibration, baseline examples, or reference samples illustrating what each score level represents. Moreover, standard reliability checks such as inter-annotator agreement and annotator-population details are entirely missing.
-	The claim of convergence acceleration in Section 4.3 might be misleading. It is not fair to consider a VQA and a narrative as a same single instance in informational v iew, since the latter is essentially aggregation of multiple VQA annotations. Also, from computational resource point of view narratives cost more tokens, so it might not be entirely accurate to claim faster convergence just based on the number of optimization steps.
-	The ablation study is missing; we ought to know at least what's the impact of each QBP and QBC is in isolation.

[1] PromptCap: Prompt-Guided Task-Aware Image Captioning (ICCV 2023)

[2] Generating Question Relevant Captions to Aid Visual Question Answering (ACL 2019)

[3] Questioning, Answering, and Captioning for Zero-Shot Detailed Image Caption (ACCV workshop 2024)

[4] Customized Image Narrative Generation via Interactive Visual Question Generation and Answering (CVPR 2018)

[5] VAGUE: Visual Contexts Clarify Ambiguous Expressions (ICCV 2025)

**Questions:**

-	I am a bit confused on the baseline for experiment in Section 4.1: Is the Qwen3-8B baseline also fine-tuned with the same dataset as TIR? If then, it should be notified otherwise (e.g. Text-Only RL) to differentiate it from vanilla Qwen3-8B.
-	Is there a validation process for Gemini’s aptness as classifier for algorithmic friendliness in Section 4.2? Without validation, we do not know how correct the friendliness score label Gemini generated is.

---

### Official Review · Reviewer_7Put · 2025-10-29

**Soundness:** 2
**Presentation:** 2
**Contribution:** 2
**Rating:** 2
**Confidence:** 4

**Summary:**

This paper introduces a novel data-centric framework for Video Question Answering (VideoQA) that addresses the fundamental limitation of "bag-of-facts" supervision in existing datasets. The authors propose two complementary strategies: Question-Based Paraphrasing (QBP), which synthesizes multiple question-answer pairs into holistic narrative paragraphs to capture event structure, and Question-Based Captioning (QBC), which generates fine-grained visual rationales to provide instance-level grounding. Through extensive experiments on STAR and NExT-QA benchmarks, the method demonstrates significant performance improvements, achieving new state-of-the-art results while also enhancing cross-dataset generalization and accelerating convergence.

**Strengths:**

1）	The paper identifies a fundamental limitation in current VideoQA datasets—the lack of narrative coherence and visual grounding—and proposes an elegant solution using synthetic data generation. The dual-strategy approach (QBP for global narrative, QBC for local grounding) is both innovative and well-justified.

2）	The method achieves significant and consistent improvements across multiple model architectures and scales. The gains of up to +5.0% on STAR and a new SOTA of 80.8% on NExT-QA are impressive and clearly demonstrate the effectiveness of the proposed supervision paradigm.

**Weaknesses:**

1）	The quality of QBP and QBC depends heavily on the capabilities of the underlying LLMs/MLLMs (e.g., GPT-4, Qwen-VL). While the human evaluation validates the output quality, the approach inherits the limitations and potential biases of these foundation models. Prompt Engineering cannot fully address the challenges within the models.

2）	Although the paper includes a qualitative error analysis, it could benefit from a more systematic categorization and quantification of error types. For instance, how often do temporal ordering errors in QBP or "justified fabrication" in QBC occur, and what is their impact on downstream performance?

3）	While the paper positions itself well against prior work, it could more directly compare against other LLM-based data augmentation techniques for VideoQA to better isolate the contribution of narrative structure versus simply having more data.

4）	The experiments focus on two primary benchmarks (STAR and NExT-QA). Including results on more diverse video domains (e.g., movies, egocentric videos) would strengthen the claim of generalizability.

**Questions:**

See the weakness.

---

### Meta-Review · Area_Chair_bZnB · 2026-01-06

**Summary:**

All reviewers unanimously recommend rejection due to marginal novelty and unconvincing experiments.
There was no rebuttal. No basis to accept.

**Reviewer Concerns:**

No rebuttal from the authors.

**Reviewer Scores:**

Due to the absence of rebuttal, no basis to change their scores.

---

### Decision · Program_Chairs · 2026-01-26

Reject